# Towards the Elucidation of the Pharmacokinetics of Voriconazole: A Quantitative Characterization of Its Metabolism

**DOI:** 10.3390/pharmaceutics14030477

**Published:** 2022-02-22

**Authors:** Josefine Schulz, Antonia Thomas, Ayatallah Saleh, Gerd Mikus, Charlotte Kloft, Robin Michelet

**Affiliations:** 1Department of Clinical Pharmacy & Biochemistry, Institute of Pharmacy, Freie Universitaet Berlin, Kelchstraße 31, 12169 Berlin, Germany; josefine.schulz@fu-berlin.de (J.S.); antonia.thomas@fu-berlin.de (A.T.); ayatallah.saleh@fu-berlin.de (A.S.); gerd.mikus@fu-berlin.de (G.M.); charlotte.kloft@fu-berlin.de (C.K.); 2Graduate Research Training Program PharMetrX, 12169 Berlin/Potsdam, Germany; 3Department of Clinical Pharmacology and Pharmacoepidemiology, University Hospital Heidelberg, Im Neuenheimer Feld 410, 69120 Heidelberg, Germany

**Keywords:** voriconazole, drug metabolism, pharmacokinetics, CYP P450 enzymes, CYP inhibition, intersystem extrapolation factors, time-dependent inhibition

## Abstract

The small-molecule drug voriconazole (VRC) shows a complex and not yet fully understood metabolism. Consequently, its in vivo pharmacokinetics are challenging to predict, leading to therapy failures or adverse events. Thus, a quantitative in vitro characterization of the metabolism and inhibition properties of VRC for human CYP enzymes was aimed for. The Michaelis–Menten kinetics of voriconazole *N*-oxide (NO) formation, the major circulating metabolite, by CYP2C19, CYP2C9 and CYP3A4, was determined in incubations of human recombinant CYP enzymes and liver and intestine microsomes. The contribution of the individual enzymes to NO formation was 63.1% CYP2C19, 13.4% CYP2C9 and 29.5% CYP3A4 as determined by specific CYP inhibition in microsomes and intersystem extrapolation factors. The type of inhibition and inhibitory potential of VRC, NO and hydroxyvoriconazole (OH–VRC), emerging to be formed independently of CYP enzymes, were evaluated by their effects on CYP marker reactions. Time-independent inhibition by VRC, NO and OH–VRC was observed on all three enzymes with NO being the weakest and VRC and OH–VRC being comparably strong inhibitors of CYP2C9 and CYP3A4. CYP2C19 was significantly inhibited by VRC only. Overall, the quantitative in vitro evaluations of the metabolism contributed to the elucidation of the pharmacokinetics of VRC and provided a basis for physiologically-based pharmacokinetic modeling and thus VRC treatment optimization.

## 1. Introduction

The small-molecule drug voriconazole (VRC, Figure 1), a broad-spectrum, triazole antifungal agent inhibiting the ergosterol biosynthesis of the fungal cell wall [1], is listed by the World Health Organization as an essential medicine for adults and children [2,3]. VRC is used worldwide in first-line therapy of invasive fungal infections such as aspergillosis. Further areas of application include the treatment of candidaemia and infections caused by *Scedosporium* sp. And *Fusarium* sp., as well as prophylaxis in patients with haematopoietic stem-cell transplantation [4]. The prevalence of these infections has increased in recent years concurrently with the expanding susceptible patient population, e.g., patients receiving immunosuppressants [5,6,7]. In the context of rising resistance to antifungal agents, responsible stewardship of existing antifungals, such as VRC, is needed more than ever [8,9,10,11]. Although VRC has been marketed since 2002 in the European Union and 2003 in the United States of America, its pharmacokinetics (PK) is not yet fully understood [12,13]. In particular, the large intra- and interindividual variabilities observed in clinical practice challenge the current VRC standard dosing regimen for ensuring a safe and effective therapy [14,15,16,17,18,19,20,21,22,23,24].

One important step towards rational, patient-individual dosing and the elucidation of VRC’s PK is a deeper understanding of its metabolism. VRC shows a complex metabolism, with 98% of the dose undergoing metabolic transformations and only 2% excreted unchanged in urine [14,25,26]. The main metabolite circulating in plasma is voriconazole *N*-oxide (NO, Figure 1), which does not contribute to the antifungal activity of VRC but is suspected of causing adverse events [4,14,27,28,29,30]. The enzymes involved in its formation encompass the cytochrome P450 (CYP) enzymes 2C19, 3A4 and 2C9, as well as, subordinately, flavin-containing monooxygenases (FMO) [14,31,32,33]. Moreover, the individual contribution of the respective enzymes to VRC *N*-oxidation has been discussed; the involvement of CYP2C9 especially has been questioned [33,34]. On the contrary, the major influence of the polymorphic CYP2C19 has been demonstrated in many clinical trials [18,23,35,36,37,38,39,40,41]. Numerous other metabolites, with lower proportions in VRC’s metabolism, have been mainly detected in urine, e.g., hydroxyvoriconazole (OH–VRC, Figure 1) and dihydroxyvoriconazole [14,23,24,25,42]; nevertheless, the pathway of their formation remains unknown. Furthermore, VRC is considered a potent CYP inhibitor, having effects on CYP2C19, CYP2C9, CYP3A and CYP2B6 [43,44,45] and therefore the potential to cause a variety of drug–drug interactions [4,14,46]. As a consequence, the nonlinear PK of VRC has been assumed to be caused by saturable metabolic processes and potentially by auto-inhibition mechanisms [42,47].

Future perspectives on VRC’s dose optimization as well as its individualization lie within in silico approaches such as physiologically-based pharmacokinetic (PBPK) modeling. PBPK modeling enables the simultaneous integration of a wide range of human (patho)physiological parameters and the physicochemical properties of a drug to predict the overall systemic or tissue exposure of the drug over time [48]. As a consequence, observations on one population, e.g., adults, can be extrapolated to another, e.g., children, to guide safe and effective VRC treatment [49]. Nevertheless, an essential prerequisite for reliable predictions is the quality of the quantitative data, as input parameters, that the model is based on.

Thus, we aimed at: (i) performing a comprehensive quantification of VRC’s metabolism to NO in different in vitro enzymatic systems, (ii) evaluating the contribution of different CYP enzymes to NO formation and (iii) investigating the metabolic fate of NO and OH–VRC. Furthermore, (iv) the inhibitory potential of VRC, NO and OH–VRC, as well as the mechanism on the CYP enzymes 2C19, 2C9 and 3A4, were assessed to explore the causes of the observed nonlinear PK in humans. Overall, a coherent, quantitative framework of VRC’s metabolism was aimed for, to contribute to the further elucidation of VRC’s PK and provide a solid database of input parameters essential in PBPK modeling.

## 2. Materials and Methods

### 2.1. Chemicals, Drugs and Enzyme Systems

VRC, NO and OH–VRC reference standards were purchased from Toronto Research Chemicals (Toronto, ON, Canada). The marker substrates S-mephenytoin, diclofenac (disodium salt) and midazolam (1 mg/mL solution in methanol) were purchased from Cayman Chemical (Ann Arbor, MI, USA), Sigma-Aldrich (Saint Louis, MI, USA) and Merck KGaA (Darmstadt, Germany), respectively. The CYP inhibitors loratadine and ketoconazole were purchased from Tokyo Chemical Industry (Tokyo, Japan), and sulfaphenazole was purchased from Cayman Chemical (Ann Arbor, MI, USA). Pooled human liver microsomes (HLM); recombinant human cytochrome P450 enzymes (rhCYP) 2C19, 2C9 and 3A4; and pooled human intestine microsomes (HIM) were purchased from Corning Inc. (New York, NY, USA). The energy providing NADPH re-generating system was bought as solutions A (26 mM NADP^+^, 66 mM glucose-6-phosphate, 66 mM magnesium chloride) and B (40 U/mL glucose-6-phosphate dehydrogenase in 5 mM sodium citrate) from Corning Inc. (New York, NY, USA). Methanol (MeOH) was purchased from Fisher Scientific (Schwerte, Germany), and ultrapure (UP) water was provided by a LaboStar™ 2-DI/UV device (Evoqua Water Technologies, Günzburg, Germany). Incubation (TRIS–HCl, pH 7.5, 250 mM) and homogenization buffer (potassium phosphate buffer, pH 7.4, 50 mM) were freshly prepared from tris(hydroxymethyl)aminomethane (TRIS, Thermo Fisher Scientific Inc., Rochester, New York, NY, USA) and hydrochloric acid 37% (Merck KGaA, Darmstadt, Germany) and dipotassium hydrogen phosphate trihydrate and potassium dihydrogen phosphate (both Merck KGaA, Darmstadt, Germany), respectively.

### 2.2. Incubation Conditions and Assay Procedure

All incubations were performed at 37 °C and pH 7.5 in TRIS buffer (50 mM) under gentle, constant shaking in an Eppendorf Thermomixer^®^. Final incubations (100 µL) consisted of the respective enzyme system, the NADPH re-generating system (1.3 mM NADP^+^, 3.3 mM magnesium chloride, 3.3 mM glucose-6-phosphate and 0.4 U/mL glucose-6-phosphate dehydrogenase), the respective substrate, and, where applicable, the respective inhibitor. The reaction time started with the addition of NADPH re-generating system A or the enzyme working solution. The enzymatic reaction was stopped by the addition of 20 µL incubation mixture to 80 µL dry-ice-cooled methanol. All samples were vortex-mixed immediately and stored at −80 °C until analysis.

### 2.3. Bioanalysis

Quantification of NO [50], 4-hydroxymephenytoin, 4-hydroxydiclofenac and 1-hydroxymidazolam was performed using a liquid-chromatography tandem mass spectrometry (LC-MS/MS) assay. All samples were centrifuged for 15 min at 14,000× *g* at 4 °C and a 20 µL aliquot of the supernatant was transferred to 20 µL of internal standard solution (2 or 10 ng/mL diazepam in UP water). The LC-MS/MS system (Agilent Technologies, Waldbronn, Germany) used for quantification combined an Agilent 1290 Infinity II LC system—including a multisampler (G7167B), a high-speed pump (G7120A) and a multicolumn thermostat (G7116B)—with a triple quadrupole MS/MS system (G6495A). A gradient method applying MeOH and UP water, both containing 0.1% formic acid (*v*/*v*), ensured the chromatographic separation and elution on an InfinityLab Poroshell 120 Phenyl Hexyl column (RP, 2.1 × 100 mm, 2.7 µm, Agilent Technologies, Waldbronn, Germany). The electrospray ionization source of the MS was operated in positive ionization mode and ion acquisition was performed by dynamic multiple reaction monitoring. Transitions monitored for quantification were *m*/*z* 366 → 224 for NO, *m*/*z* 235→ 150 for 4-hydroxymephenytoin, *m*/*z* 312 → 230 for 4-hydroxydiclofenac, *m*/*z* 342 → 324 for 1-hydroxymidazolam and *m*/*z* 285 → 193 for the internal standard (diazepam).

Concentrations for calibration samples were adapted to the respective experiments, and consisted at all times of a blank, a zero and eight calibrators and ranged from 0.1 to 500 ng/mL for NO, 0.5 to 400 ng/mL for 4-hydroxymephenytoin, from 1 to 1000 ng/mL for 4-hydroxydiclofenac and from 0.1 to 1000 ng/mL for 1-hydroxymidazolam, respectively. In all runs, a minimum of three concentration levels of quality control samples (low, medium and high) were included and runs only accepted if a minimum of 67% of QC samples were within ±15% of their nominal concentration with a minimum of 50% accurate QC samples per level [51].

### 2.4. Voriconazole and Its Metabolites as Substrates

As an essential prerequisite, the linearity of metabolite formation with regard to reaction time and enzyme concentration, as well as the absence of substrate depletion (<10%), were investigated. As a result, the linearity of NO formation with regard to time was demonstrated in HLM, HIM and rhCYP, as well as regarding enzyme concentration in HLM and rhCYP. The enzyme concentrations and reaction times applied during all kinetic investigations ensured the absence of VRC depletion. During these pre-investigations, OH–VRC formation was not observed, indicating a CYP- and FMO-independent pathway of formation and resulting in a focus on NO in the following kinetic investigations.

#### 2.4.1. Michaelis–Menten Kinetics of Voriconazole *N*-oxidation

For the assessment of kinetic parameters, HLM (0.2 mg/mL) were incubated with VRC (0.5, 2, 3, 4, 5, 8, 10, 15, 20, 30, 60 and 100 µM) for 10, 20 and 30 min (*n* = 3). Investigations in recombinant systems were performed for 15 and 25 min at enzyme concentrations of 5 (*n* = 3) and 15 pmol/mL (*n* = 6) rhCYP2C19, 20 (*n* = 3) and 40 pmol/mL (*n* = 6) rhCYP3A4 and 100 pmol/mL (*n* = 5) rhCYP2C9, respectively. VRC concentrations of 0.5, 1, 2, 3, 4, 8, 30 and 100 µM were used. VRC *N*-oxidation in HIM (0.5 mg/mL) was assessed for 5, 15 and 25 min applying VRC substrate concentrations of 0.5, 1, 2, 3, 4, 5, 8, 10, 15, 30, 60 and 100 µM (*n* = 3).

The reaction velocity (V, in pmol/min·mg for HLM and HIM and pmol/min·pmol for rhCYP) was determined for each sample as the concentration of the metabolite (C_Met_, in pmol/mL) divided by the respective reaction time (t_reac._, in min) and concentration of HLM, rhCYP or HIM (C_Enyzme_, in mg/mL for HLM and HIM and pmol rhCYP/mL for rhCYP, Equation (1)).
(1)$$V\;=\frac{C_{\mathrm{Met}}}{C_{\mathrm{Enzyme}}\cdot t_{\mathrm{reac}.}}$$

The reaction velocity of metabolite formation (V) was analyzed as a function of the added substrate concentration ([S], in µM) using the Michaelis-Menten equation (Equation (2)). Kinetic parameters, i.e., the Michaelis–Menten constant (K_M_, in µM) and maximum reaction velocity (V_max_, in pmol/min·mg for HLM and HIM and pmol/min·pmol for rhCYP), were estimated by nonlinear regression using the “nls” function in R and R Studio^®^ (version 3.6.3, Vienna, Austria). The intrinsic clearance value (CL_int_, in µL/min·mg for HLM and HIM and µL/min·pmol for rhCYP) was determined as the respective ratio of V_max_ and K_M_.
(2)$$V\;=V_{\mathrm{Max}}\cdot \frac{\left[S\right]}{K_{M}+\left[S\right]}$$

#### 2.4.2. Contribution of Individual CYP Enzymes to Voriconazole *N*-oxidation

To quantify the contribution of the different CYP enzymes in VRC *N*-oxidation, two different experimental approaches were applied. In the first approach, the specific CYP inhibitors, loratadine (10 µM, CYP2C19), sulfaphenazole (10 µM, CYP2C9) and ketoconazole (0.1 µM, CYP3A4), as well as a mixture of the three, were added to VRC (0.5, 1, 2, 3 µM) incubations in HLM (*n* = 3, t_reac_.= 15 and 25 min) [52,53,54,55,56,57,58]. The reaction velocities at the respective substrate concentrations in the presence and absence of the inhibitor were used to assess the contribution to NO formation. The specificity of the enzyme inhibition by loratadine, sulfaphenazole and ketoconazole was tested using marker reactions for CYP2C19, CYP2C9 and CYP3A4 (Appendix A).

In the second approach, extrapolation from recombinant human CYP enzymes was performed using intersystem extrapolation factors (ISEF) [59,60]. The ISEF was determined by two different approaches: first, as the V_max_ ratio of the respective marker reaction in HLM and rhCYP corrected for CYP abundance in HLM (P450i abundance_HLM_, in pmol/mg), Equation (3); second, as the ratio of CL_int_ of the respective marker reaction in HLM and rhCYP corrected for CYP abundance in HLM (Equation (4)) (Appendix A).
(3)$${\mathrm{ISEF}}_{V_{\max},\mathrm{CYPi}}=\frac{V_{\max,\;\mathrm{MR},\mathrm{HLM}}}{V_{\max,\mathrm{MR},\mathrm{rhCYP}}\cdot P450{i\;\mathrm{abundance}}_{\mathrm{HLM}}}$$
(4)$${\mathrm{ISEF}}_{{\mathrm{CL}}_{\mathrm{int}},\mathrm{CYPi}}=\frac{{\mathrm{CL}}_{\mathrm{int},\;\mathrm{MR},\mathrm{HLM}}}{{\mathrm{CL}}_{\mathrm{int},\mathrm{MR},\mathrm{rhCYP}}\cdot P450{i\;\mathrm{abundance}}_{\mathrm{HLM}}}$$

Individual CYP enzyme contributions to VRC *N*-oxidation were calculated by using CL_int_ determined in rhCYP (CL_int,rhCYPi_) and converting them to the respective CL_int_ in HLM (CL_int,CYPi,HLM_) by taking into account the individual CYP abundances as well as the respective ISEF (Equation (5)). The individual clearance of each enzyme was related to the overall clearance observed in HLM (CL_int,HLM_, Equation (6)).
(5)$${\mathrm{CL}}_{\mathrm{int},\mathrm{CYPi},\mathrm{HLM}}={\mathrm{ISEF}}_{\mathrm{CYPi}}\cdot {\mathrm{CL}}_{\mathrm{int},\mathrm{rhCYPi}}\cdot P450i\;\mathrm{abundance}$$
(6)$${\mathrm{Contribution}}_{\mathrm{CYPi}},\;\%=\frac{{\mathrm{CL}}_{\mathrm{int},\mathrm{CYPi},\mathrm{HLM}}}{{\mathrm{CL}}_{\mathrm{int},\mathrm{HLM}}}\cdot 100\%$$

#### 2.4.3. In Vitro In Vivo Extrapolation

CL_int_ values determined using in vitro metabolic systems were extrapolated to in vivo hepatic intrinsic clearances (CL_int,hepatic,invivo_, in mL/min) by multiplication with the microsomal protein per gram liver (MPPGL, in mg/g) and mean human liver mass (in g, Equation (7)) [60].
(7)$${\mathrm{CL}}_{\mathrm{int},\mathrm{hepatic},\mathrm{invivo}}=\mathrm{MPPGL}\cdot \mathrm{liver}\;\mathrm{mass}\cdot {\mathrm{CL}}_{\mathrm{int},\mathrm{VRC},\;\mathrm{HLM}\;}$$

Furthermore, by applying the well-stirred liver model [61,62], which takes hepatic blood flow (Q_H_, in mL/min), the fraction unbound in plasma (fu_p_) and the blood-plasma-ratio (BP) into account, an in vivo hepatic plasma clearance (CL_hepatic_, in mL/min) was calculated (Equation (8)).
(8)$${\mathrm{CL}}_{\mathrm{hepatic}}=\frac{Q_{H}\cdot {\mathrm{fu}}_{p}/\mathrm{BP}\cdot {\mathrm{CL}}_{\mathrm{int},\mathrm{hepatic},\mathrm{invivo}}}{Q_{H}+{\mathrm{fu}}_{p}/\mathrm{BP}\cdot {\mathrm{CL}}_{\mathrm{int},\mathrm{hepatic},\mathrm{invivo}}}$$

Respective values for MPPGL, liver mass, Q_H_, fu_P_ and BP were taken from literature and are collected in Appendix A.

#### 2.4.4. Metabolic Stability of Voriconazole *N*-oxide and Hydroxyvoriconazole

For investigations on NO’s and OH–VRC’s stability in HLM incubations, continued metabolism to secondary VRC metabolites and degradation, both substances were directly incubated with HLM. The depletion of NO and OH–VRC at low (0.137 µM) and high (1.10 µM) concentrations was assessed over time in incubations containing 0.2 mg/mL HLM. After 5, 15, 30 and 60 min, samples were taken and the respective concentration of NO or OH–VRC was determined (*n* = 2). An incubation without HLM served as a control (*n* = 1).

### 2.5. Voriconazole and Its Metabolites as Inhibitors

To assess VRC, NO and OH–VRC as CYP inhibitors, marker reactions were monitored for the activities of CYP2C19, CYP2C9 and CYP3A4, respectively (Appendix A). The concentration at which VRC, NO and OH–VRC displayed their half maximal inhibitory effect (IC_50_, in µM) was determined at substrate concentrations close to the K_M_ of the marker reaction (55.0 µM S-mephenytoin, 4.73 µM diclofenac and 4.60 µM midazolam). Five inhibitor concentrations ranging from 0 to 100 µM for VRC and from 0 to 34.2 µM for NO and OH–VRC were applied (*n* = 2). Additionally, the same procedure was carried out, including a 30 min pre-incubation in the presence or absence of NADPH, to detect time-dependent inhibition (IC_50_ shift). Remaining activity (A, %) was calculated as the ratio of reaction velocities in the inhibited and non-inhibited incubations. IC_50_ and the steepness of the inhibition curve (H) were estimated by fitting the four-parameter inhibition model to the experimental data (Equation (9)). Maximum (A_max_) and baseline activity (A_0_) were fixed to 100 and 0, respectively. In case of time-dependent inhibition, IC_50_ values were expected to shift to lower values, and shifts of >1.5-fold have been described previously to be physiologically relevant [63].
(9)$$A,{\%=A}_{0}+\frac{\left(A_{\max}-A_{0}\right)}{1+{\left(\frac{\log\left(C_{\mathrm{Inhib}}\right)}{\log\left({\mathrm{IC}}_{50}\right)}\right)}^{H}}=\frac{100}{1+{\left(\frac{\log\left(C_{\mathrm{Inhib}}\right)}{\log\left({\mathrm{IC}}_{50}\right)}\right)}^{H}}$$

The type of reversible inhibition as well as the inhibitory constant (K_i_) were assessed by incubating the marker substrates S-mephenytoin (9.16, 13.7, 18.3, 55.0, 91.6 and 458 µM), diclofenac (1.69, 3.38, 4.73, 6.75, 27.0 and 169 µM) and midazolam (0.767, 1.53, 3.07, 4.60, 9.21 and 30.7 µM) in the presence of VRC (0, 0.10, 1.0, 5.0, 10.0 and 100 µM), NO and OH–VRC (both 0, 0.0958, 0.958, 3.83, 9.58 and 34.2 µM) (*n* = 2). Reaction velocities were investigated in function of the substrate ([S]) and inhibitor ([I]) concentration to determine K_M_, V_max_ and K_i_ of the reaction. The obtained data were used to fit the three main models of reversible inhibition, i.e., competitive (Equation (10)), noncompetitive (Equation (11)) and uncompetitive (Equation (12)) using the “nls” function in R. The type of inhibition was determined according to the best model fit assessed with Akaike’s information criteria (AIC) [64]. A model was deemed to be significantly better when the difference in AIC exceeded 2.
(10)$$V=\frac{V_{\max}}{1+\frac{K_{M}}{\left[S\right]}\left(1+\frac{\left[I\right]}{K_{i}}\right)}$$
(11)$$V=\frac{V_{\max}}{\left(1+\frac{K_{M}}{\left[S\right]}\right)\left(1+\frac{\left[I\right]}{K_{i}}\right)}$$
(12)$$V=\frac{V_{\max}}{\left(1+\frac{K_{M}}{\left[S\right]}+\frac{\left[I\right]}{K_{i}}\right)}$$

## 3. Results

### 3.1. Michaelis-Menten Kinetics of Voriconazole N-oxidation

The kinetics of VRC *N*-oxidation in all enzymatic systems were characterized by Michaelis-Menten kinetics (Figure 2). The estimated kinetic parameters, i.e., K_M_ and V_max_, are summarized in Table 1.

In HLM, this resulted in a CL_int_ of VRC *N*-oxidation of 8.76 µL/min·mg (7.63–10.2 µL/min·mg). HIM also relevantly metabolized VRC to NO. The reaction kinetics were characterized by a CL_int_ of 1.52 µL/min·mg (1.27–1.86 µL/min·mg) and were thus 5.76-fold lower than CL_int_ observed in HLM.

In recombinant enzymes, rhCYP2C19 showed the highest VRC turnover indicated by a CL_int_ of 1.25 µL/min·pmol (0.857–2.05 µL/min·pmol). Kinetics for rhCYP2C9 were determined to have a CL_int_ of 0.00173 µL/min·pmol (0.00138–0.00224 µL/min·pmol), which was 723-fold lower than that in rhCYP2C19. Lastly, in rhCYP3A4, Michaelis–Menten kinetics of VRC *N*-oxidation were described by a CL_int_ of 0.00742 µL/min·pmol (0.00523–0.0115 µL/min·pmol), which was 168-fold lower and 4.29-fold higher than the CL_int_ observed in rhCYP2C19 and rhCYP2C9, respectively.

### 3.2. Contribution of Individual CYP Enzymes to Voriconazole N-oxidation

Different methods can be applied to determine the contribution of CYP enzymes to the overall formation of a certain metabolite. The two approaches implemented for VRC comprised: (i) the individual inhibition of CYP enzymes in HLM and (ii) the extrapolation from recombinant enzymes using ISEF.

(i) The effect of the inhibitors loratadine (CYP2C19), ketoconazole (CYP3A4) and sulfaphenazole (CYP2C9) on the reaction velocities of VRC *N*-oxidation were evaluated using four VRC concentrations ≤ K_M_ (2.98 µM). CYP2C19 had the largest share in VRC *N*-oxidation with 62%, followed by CYP3A4 with 48% and CYP2C9 with 36% (Table 2). Evidently, an inhibition of >100% in sum was implausible and indicated that the inhibitors were not sufficiently specific for the respective enzymes. Thus, the three inhibitors were tested on marker reactions catalyzed by mainly one enzyme (Appendix A). In brief, loratadine inhibited the formation of 4-hydroxymephenytoin by CYP2C19 reliably with less than 5% of the metabolism remaining. Additionally, it had no effect on the 1-hydroxylation of midazolam by CYP3A4, with reaction velocities remaining unchanged in its presence. However, there was a cross-reaction with the 4-hydroxylation of diclofenac by CYP2C9, as only 42% of the metabolism remained. Similar observations were made for sulfaphenazole which inhibited CYP2C9 almost completely (6.3% of the metabolism remaining) and CYP3A4 not at all with 97% of the metabolism remaining. Yet it also inhibited CYP2C19, with reaction velocities of 77% compared to the uninhibited control. Lastly, ketoconazole had minor effects on CYP2C19 and CYP2C9, with 92% and 85% remaining metabolism, respectively, but inhibited the 1-hydroxylation of midazolam by CYP3A4, with 37% persisting reaction velocity (Appendix A). Overall, the observed effect of the three combined inhibitors on VRC *N*-oxidation might be the most conclusive, as the specificity was of less importance. Here, 89% of the metabolism was inhibited, indicating that the most relevant enzymes were detected—in particular considering that CYP3A4 inhibition was insufficient, potentially due to a too low ketoconazole concentration.

(ii) Incubations of rhCYP2C19, rhCYP2C9 and rhCYP3A4 all revealed distinct VRC *N*-oxidation. To compare their clearance to the overall observed CL_int_ in HLM, the respective CYP abundance in HLM was considered, as well as an ISEF to account for activity differences in the two metabolic systems. Michaelis-Menten kinetics of marker reactions for CYP2C19, CYP2C9 and CYP3A4 were evaluated under laboratory individual conditions in HLM and rhCYP (Appendix A). When V_max_ was used for ISEF determination, factors resulted in 0.573, 2.06 and 3.60 for CYP2C19, CYP2C9 and CYP3A4, respectively. Consequently, extrapolated CL_int,CYPi,HLM_ yielded 7.89, 0.218 and 2.48 µL/min·mg and respective contributions of 90.0%, 2.49% and 28.4% for CYP2C19, CYP2C9 and CYP3A4 in VRC *N*-oxidation, respectively. Furthermore, when ISEF was based on the CL_int_ of the marker reactions, it resulted in factors of 0.239, 1.13 and 1.59 and hence in extrapolated CL_int,CYPi,HLM_ values for VRC *N*-oxidation of 3.30, 0.120 and 1.10 µL/min·mg for CYP2C19, CYP2C9 and CYP3A4, respectively. Based on this, enzyme contributions of 37.6% (CYP2C19), 1.37% (CYP2C9) and 12.5% (CYP3A4) were obtained.

Considering a mean of both ISEF approaches and the inhibition investigation in HLM, contributions of 63.1% CYP2C19, 13.4% CYP2C9 and 29.5% CYP3A4 to the hepatic *N*-oxidation of VRC were determined.

### 3.3. In Vitro In Vivo Extrapolation

In vitro in vivo extrapolation (IVIVE) from HLM resulted in a CL_int,hepatic,invivo_ of 540 mL/min (32.4 L/h), and applying the well-stirred liver model resulted in a CL_hepatic_ of 227 mL/min (13.6 L/h). When individual CL_int_ values of rhCYP were summed up, CL_int,hepatic,invivo_ values of 652 and 278 mL/min (39.1 and 16.7 L/h) were obtained, depending on whether ISEF_Vmax_ or ISEF_CLint_ was applied, respectively. Consequently, CL_hepatic_ values of 266 and 127 mL/min (16.0 and 7.62 L/h) were determined using ISEF_Vmax_ or ISEF_CLint_, respectively.

### 3.4. Metabolic Stability of Voriconazole N-oxide and Hydroxyvoriconazole

NO and OH–VRC showed no depletion over time when incubated directly with HLM. After 60 min at the low NO concentration, 110% and 105% NO compared to a control incubation without HLM were present in the two replicates. At the high NO concentration, 96.9% and 109% NO compared to the control were determined. The same was observed for OH–VRC, which yielded a recovery of 92.5% and 108% at the low, and 94.7% at the high, OH–VRC concentration after 60 min. Additionally, no trend, i.e., decreasing concentration over time, was observable, neither for the HLM containing incubation nor for the control (Figure 3).

### 3.5. Voriconazole and Its Metabolites as Inhibitors

The inhibitory potential of VRC, NO and OH–VRC, assessed as IC_50_, differed for the three enzymes. The metabolism of S-mephenytoin by CYP2C19 was most affected by VRC, with an IC_50_ of 3.72 µM (95% confidence interval: 2.85–4.78 µM). NO and OH–VRC showed minor inhibition of CYP2C19, resulting in IC_50_ values of 288 µM (65.0–31623 µM) and 41.7 µM (26.9–89.1 µM), respectively. Thus, estimated values exceeded the experimentally applied inhibitor concentrations and hence precluded a precise evaluation of IC_50_. On CYP2C9, OH–VRC had the largest influence on the formation of 4-hydroxydiclofenac, with an IC_50_ of 3.67 µM (3.16–4.26 µM); followed by VRC, with an IC_50_ of 4.17 µM (2.54–6.51 µM); and NO, with 13.4 µM (9.90–19.1 µM). The strongest inhibition by all substances was caused on the 1-hydroxylation of midazolam by CYP3A4, which resulted in IC_50_ of 1.02 µM (0.796–1.27 µM) for OH–VRC, 1.76 µM (1.26–2.36 µM) for VRC and 4.48 µM (3.78–5.29 µM) for NO (Figure 4).

Time-dependent inhibition was investigated by determining the influence of a 30 min pre-incubation of the inhibitor and HLM in the presence and absence of the NADPH re-generating system on the IC_50_. In two cases an IC_50_ shift of >1.5-fold was observed: for the effect of OH–VRC on CYP3A4 in the absence of NADPH and for NO on CYP3A4 in the presence of NADPH (Table 3). However, in addition to the defined threshold, further aspects must be considered to define time-dependent inhibition. First, confidence intervals of the IC_50_ estimations were relevantly overlapping. Second, in the case of the effect of OH–VRC on CYP3A4 the results would be expected to be reproducible also in the presence of NADPH if no depletion of OH–VRC occurred. Both conditions were not met; thus, it was concluded that neither VRC nor its two metabolites caused time-dependent inhibition (Appendix A).

As no time-dependent inhibition was observed, the type of reversible inhibition including the respective K_i_ was evaluated. The inhibition of CYP2C19 by VRC and OH–VRC was determined as a competitive inhibition, with K_i_ values of 1.90 µM (95% confidence interval: 1.70–2.12 µM) and 11.6 µM (9.65–14.0 µM), respectively. Inhibition by NO was noncompetitive, with a K_i_ of 58.6 µM (46.8–75.2 µM), and was low overall, with the estimation exceeding the experimentally applied concentrations. Therefore, the assessment of NO has to be interpreted with caution. All three substances inhibited CYP2C9 in a competitive manner and yielded K_i_ values of 2.57 µM (2.16–3.14 µM) for VRC, 5.47 µM (4.32–7.00 µM) for NO and 2.80 µM (2.20–3.61 µM) for OH–VRC. The inhibition of VRC, NO and OH–VRC on CYP3A4 was noncompetitive and resulted in K_i_ values of 2.75 µM (2.35–3.22 µM), 5.24 µM (4.68–5.86 µM) and 2.53 µM (2.24–2.87 µM), respectively (Figure 5). The simultaneously estimated K_M_ and V_max_ values are presented in Appendix A.

Overall, K_i_ investigations confirmed observations already made during the IC_50_ assessment. NO was the least potent CYP inhibitor in all cases, with 31-, 2.1- and 1.9-fold lower effects on CYP2C19, CYP2C9 and CYP3A4, respectively, than VRC itself. OH–VRC also had a minor effect on CYP2C19, with a 6.1-fold lower inhibitory potential compared to VRC. Yet, OH–VRC was a strong inhibitor, similar in strength to VRC, when CYP2C9 and CYP3A4 were investigated with its effects being 1.1- and 0.92-fold as pronounced as that of VRC, respectively.

## 4. Discussion

In this work, a coherent, quantitative in vitro characterization of VRC’s metabolism was presented to contribute to the elucidation of its complex PK and provide reliable in vitro data for PBPK modeling to pave the way towards individualized VRC dosing regimens. Besides the precise description of the Michaelis-Menten kinetics of VRC *N*-oxidation in HLM, rhCYP and HIM, enzyme contributions were assessed by different approaches highlighting the relevance of the chosen experimental settings. Moreover, the metabolic stability of NO and OH–VRC was demonstrated and IVIVE was performed. Finally, our work for the first time assessed the inhibition kinetics not only of VRC but also its metabolites to explore their involvement in the observed nonlinear PK of VRC.

The *N*-oxidation of VRC was demonstrated in all enzymatic systems investigated, HLM, rhCYP2C19, rhCYP2C9, rhCYP3A4 and HIM, and followed nonlinear Michaelis–Menten kinetics. Derived kinetic parameters were compared to those previously reported in literature. Overall, our study went further by performing more replicates and obtaining a larger precision [31,33]. In HLM, previously reported K_M_ and V_max_ reached values of 8.1 µM (±2.1 µM) and 9.3 pmol/min·mg (±11.1 pmol/min·mg) and were based on HLM derived from three individuals [31]. Consequently, the deviations to our results are distinct (K_M_ 2.98 µM, V_max_ 26.1 pmol/min·mg). In recombinant enzymes for CYP2C19, K_M_ and V_max_ values of 14 ± 6 µM and 0.22 ± 0.02 nmol/min·nmol [33] and 3.5 µM and 0.39 pmol/min·pmol [31] have been described, representing an 11- and 3-fold higher K_M_ and a 7- and 4-fold lower V_max_ value compared to our determinations. For rhCYP3A4, K_M_ and V_max_ of 16 ± 10 µM and 0.05 ± 0.01 nmol/min·nmol [33] and 235 µM and 0.14 pmol/min·pmol [31] were observed. In comparison, our results of a K_M_ of 1.20 µM and a V_max_ of 0.00893 pmol/min·pmol were 13- and 196-fold lower with regard to K_M_ and 6- and 16-fold lower with regard to V_max_. For rhCYP2C9, one study reported a K_M_ of 20 µM and a V_max_ of 0.056 pmol/min·pmol [31], which were 5- and 8-fold higher respectively than our determinations, while another one did not find metabolism mediated by CYP2C9 [33].

In particular, the different magnitudes of reported parameters demonstrate the necessity of carefully planned and executed in vitro experiments to generate reliable results. However, previously reported studies carried some limitations, potentially resulting in discrepancies between in vitro predicted and in vivo observed PK properties. First, kinetic investigations were performed using HLM prepared from only a few individuals. Yet, the individual predisposition for CYP enzymes is very heterogenic, as shown by Achour et al.’s meta-analysis of hepatic CYP expression [65], with variabilities exceeding 100% coefficient of variation. Higher fluctuations might even be expected if not only expression but also activity were taken into account [66]. Therefore, kinetic parameters that are to be applied and interpreted in a wider context should be based on a larger pool of liver microsomes. Thus, our research, relying on pooled microsomes from 150 donors with an equal gender distribution, offers more reliable data. Second, according to regulatory agencies, in vitro kinetic investigations should be conducted at clinically relevant concentrations [53,67]. Although there is no generally recognized PK target for VRC, a recent position paper recommends VRC minimum concentrations of at least 1–2 mg/L for an efficient and a maximum of 4.5–6 mg/L for a safe therapy [68]. This translates approximately to a target range of 2.86 to 17.2 µM VRC, a concentration range that was adequately covered by the investigations presented here (0.5–100 µM). Surpassing the upper limit to a certain extent is certainly advisable, as individuals also occasionally show unexpectedly high VRC plasma concentrations, e.g., >15 mg/L (43 µM) [69]. Yet early in vitro studies of VRC applied concentrations as high as 2500 or 5000 µM [31,33]. This is connected to the third concern: the organic solvent concentrations in incubations. Enzymes are inhibited or even inactivated by too high concentrations of methanol, acetonitrile or DMSO leading to biased kinetic determinations. Current recommendations are to keep the organic solvent concentration (*v/v*) as low as possible, with a rule-of-thumb of <1% methanol in final incubations [70,71]. All our kinetic determinations for the *N*-oxidation of VRC had a maximum of 0.5% methanol. As VRC is a lipophilic drug, it is difficult to reproduce how the solubility as well as the organic solvent limit was maintained at VRC concentrations of 2500 or 5000 µM [31,33]. Fourth, the usage of high HLM or rhCYP concentrations, e.g., 0.5 or 1 mg/mL [31,33], potentially increases the nonspecific protein binding of the substrate to the protein and thus reduces the unbound fraction of the substrate [72]. Therefore, for unbiased kinetic determinations the unbound substrate concentration should be used for determinations of Michaelis-Menten kinetics [70]. However, if no data on the fraction unbound is available, as it is the case for VRC, low HLM concentrations should be chosen, having a smaller influence on the unbound fraction—as has been shown for midazolam [73]—and thus a smaller influence on the determined kinetics. Fifth and last, the observation of metabolite formation is generally favorable compared to substrate depletion [32]. Reaction kinetics change with decreasing substrate concentrations; however, this is rarely accounted for, and a depletion of 10% to 20% is defined as negligible.

With regard to the contributions of the respective enzymes to the overall formation of NO, different approaches for the determination have been explored, yielding results with mean contributions of 63.1% for CYP2C19, 13.4% for CYP2C9 and 29.5% for CYP3A4. The determination by the application of specific CYP inhibitors in HLM was interfered by cross-inhibition of the inhibitors loratadine and sulfaphenazole on the enzymes of the same family, CYP2C19 and CYP2C9. This problem is not unknown, and even the U.S. Food and Drug Administration states that for CYP2C19 no specific inhibitor is yet available [52,67]. Additionally, the specificity is highly dependent on the concentration of the inhibitor, with higher concentrations leading to larger cross-reactions. In contrast, if inhibitor concentrations are too low, an incomplete inhibition is the consequence. In our study, we aimed for the best compromise based on data in literature. Still, ketoconazole concentrations might have been too low for a complete inhibition of CYP3A, potentially explaining the 11% remaining metabolism when a combination of all inhibitors was used. Moreover, the magnitude of inhibition has been shown to be dependent on the probe substrate for CYP2C9 and CYP2C19 [55,57], indicating different binding modalities of different substrates. Thus, controls on well-established marker reactions for specificity testing of the inhibitor also have to be considered to have some limitations. A similar argument can be raised when extrapolation from recombinant systems was applied. Here too, ISEF determination relied on marker reactions based on the assumption that substrates of the same CYP enzyme behave more or less identically. This is not always the case, as shown previously [74,75]. Nevertheless, the different results can be interpreted in context and summarized in the following major conclusions: (i) CYP2C19 played the largest role in VRC *N*-oxidation; (ii) the contribution of CYP2C9 was negligible; and (iii) besides CYP3A4, no further enzymes were contributing significantly to NO formation. While conclusion (ii) is in accordance with the reported safety of VRC in a poor metabolizer of CYP2C9 in a clinical trial [34], conclusion (iii) contrasts with investigations by Yanni et al., who found a 25% contribution of FMO [32]. Furthermore, a clinical trial by Mikus et al.—investigating potential reasons for differences in VRC clearance in CYP2C19 normal and poor metabolizers in the absence and presence of the strong CYP3A4 inhibitor ritonavir—came to the conclusion that CYP2C19 is responsible for 66% of the metabolism and CYP3A4 for 34%. In the case of CYP2C19 poor metabolizers with ritonavir treatment, the metabolism was decreased by 86% [76]. Overall, our in vitro results regarding enzyme contributions corresponded very well to those observations, potentially even explaining the remaining 14% as being CYP2C9 mediated metabolism.

The performed IVIVE additionally confirmed the validity of the generated data. In clinical studies, VRC clearances of 244 mL/min (first dose, all genotypes) [24], 420 mL/min (first dose, normal metabolizers), 194 mL/min (first dose, heterozygous normal metabolizers), 149 mL/min (first dose, poor metabolizers) [23] and 272 mL/min (multiple dose, all genotypes) [77] have been reported. Thus, the estimated hepatic clearance of 127–266 mL/min, as derived from the IVIVE presented here, represents a good approximation, considering that only one metabolic pathway, VRC *N*-oxidation, was taken into account. The extrapolation from HIM and the determination of the fraction of the intestinal (intrinsic) clearance of the total VRC metabolic clearance is a promising next step from the current work by the application of PBPK modeling.

The role of other pathways, e.g., (di-)hydroxylation of VRC at the fluoropyrimidine moiety, has been controversially discussed in literature. Although NO is the major *circulating* metabolite, the formation of (di-)hydroxy-VRC has been claimed to be the major pathway, and low plasma concentrations have been explained by a high clearance [14,23]. However, in our investigations OH–VRC was not formed in incubations of HLM, rhCYP or HIM. Additionally, the formation of dihydroxyvoriconazole, a secondary metabolite of OH–VRC, was indirectly excluded in the experiments by the observed absence of depletion of OH–VRC in HLM. These findings suggest a different pathway of formation for both metabolites.

Reaction kinetics in HIM have previously not been assessed. CL_int_ was approximately 6-fold lower than in HLM, which was probably due to the lower CYP enzyme abundance per gram of microsomal protein in the small intestine compared to the liver. For CYP2C19, CYP2C9 and CYP3A4 in the small intestine abundances of 2.1, 11 and 58 pmol/mg have been described, which were respectively 5.2-, 5.5- and 1.6-fold lower than those in the liver (Appendix A) [65,78]. Nevertheless, our results suggest that metabolic transformation by the intestine should not be ignored. This is in line with clinical trials where per oral (p.o.) and intravenous (i.v.) doses of 400 mg VRC resulted in similar maximum plasma concentrations and areas under the concentration-time-curve, but doses of 50 mg did not. Moreover, bioavailability has been observed to increase in CYP2C19 poor metabolizers (94% compared to 75% in extensive metabolizers), supporting the hypothesis of saturable metabolic processes [23]. Consequently, a reevaluation of the switch from i.v. to p.o. dosing without dose-adaptation in the recommended VRC standard dosing regimen might be advisable. For this purpose, PBPK modeling can be a beneficial tool.

In adults, VRC exhibits nonlinear PK, an observation assumed to originate in saturation or an auto-inhibition of its metabolism [42,79]. The inhibitory potential of VRC has been described thoroughly in the literature in terms of observed drug–drug interactions in vivo [4,14,46,80], as well as based on in vitro experiments [43,44,81]. Previously in vitro determined IC_50_ values for CYP2C19, CYP2C9 and CYP3A4 were <10.5 µM [43,44,81]; however, as IC_50_ determination is highly dependent on the substrate concentration used, a comparison of K_i_ values is more reliable. Here, Jeong et al. reported a K_i_ value of 2.79 µM and a competitive inhibition by VRC of CYP2C9, which is in good agreement with our findings (2.57 µM, competitive inhibition). On CYP2C19 the authors also described a competitive inhibition of CYP2C19; however, their determined K_i_ of 5.07 µM was 2.7-fold higher than our result. Lastly, on CYP3A4 Jeong et al. found a mixed competitive and noncompetitive inhibition by VRC with K_i_ values of 0.66 and 2.97 µM, respectively [43], while we demonstrated a noncompetitive inhibition only (K_i_ 2.75 µM). Deviations might arise from the different evaluation methods; Jeong et al. used a graphical tool, Dixon plots, which are based on a linear transformation of the data, while our results were based on the best nonlinear model fit and thus less error prone. Furthermore, in this case also the choice of marker reactions might influence the determined parameters [55,57].

Time-dependent inhibition of CYP2C19, CYP2C9 and CYP3A34 was not observed in our experiments, which is in line with two previously reported findings [43,44], but contrasts with another one [82]. However, the latter applied an experimental approach already critically evaluated [83]. For the inhibitory potential of NO, only two investigations of IC_50_ have been published, both resulting in different conclusions. We demonstrated a relevant inhibitory effect of NO on CYP3A4 and, although to a lesser extent, CYP2C9, but only a minor one on CYP2C19, which is in line with Giri et al. (IC_50_ values of 11.2 and 8.7 µM on CYP2C9 and CYP3A4, respectively) [84]. Hohmann et al. found a higher inhibition potential of NO on CYP2C19 (IC_50_ 40.2 µM) than CYP3A4 (IC_50_ 146 µM) [42]. OH–VRC has not been investigated previously, but in our studies partly showed a higher inhibitory potential than VRC (a 1.1-fold higher K_i_ for CYP3A4). The clinical relevance of the investigated inhibition is challenging to predict. Although plasma concentrations of OH–VRC are up to 30-fold lower than those of NO [14], local liver concentrations are of greater interest. In this context, the transport out of the hepatocytes after the formation of the metabolite in particular is crucial and needs further investigation, e.g., by applying PBPK modeling.

Indeed, for prospective predictions of VRC exposure in humans, developing a PBPK model is suggested to provide a mechanistic framework to integrate VRC’s physicochemical properties, the presented in vitro hepatic metabolism data, and other physiological processes and parameters [48,85]. The increasing availability of in silico and in vitro systems that act as easily-accessible surrogates for in vivo determinations of absorption, distribution, metabolism and excretion (ADME) processes, as well as advancements in the IVIVE techniques, are crucial reasons why PBPK modeling is becoming more appealing [86]. An IVIVE–PBPK linked model is considered a valuable tool for hypothesis testing for investigating the impact of individual, drug-related PK assumptions, which can be confirmed by existing in vivo observations. Thus, ultimately knowledge gaps in VRC’s PK can be unveiled through a learn–predict–confirm paradigm.

Furthermore, IVIVE–PBPK linked models enable the extrapolation of the PK of VRC to vulnerable patient populations, e.g., pediatrics [49], to support dosing decisions. However, as a prerequisite the metabolic and elimination pathways and the contribution of different enzymes to each pathway have to be well characterized using validated and reliable in vitro experiments [87]. For VRC, several PBPK models have been published recently, which mostly included data from in vitro metabolism investigations performed in the early phases of VRC’s marketing [82,88,89]. However, those experiments were not designed for future applications in PBPK modeling and knowledge of VRC’s PK, as well as the execution of in vitro metabolism experiments, has since increased.

Ultimately, the combination of a VRC PBPK model with a pharmacodynamic (PD) model could be used to estimate the time course of the drug response for various dosing regimens in different populations or disease states [48,90]. Simulations based on such a model could then be used to: (i) inform and qualify the model using clinically observed data, and (ii) recommend optimal dosing regimens for individual patients who have developed invasive fungal infections. Overall, this approach could translate research results from in vitro via in silico to clinical practice, supporting future therapeutic decisions.

## 5. Conclusions

In vitro investigations in HLM, rhCYP and other enzyme sources are a powerful tool for the assessment and evaluation of human metabolism. However, there is a substantial difference in the identification and basic kinetic determinations of enzymes involved in the metabolism of a (new) drug and a profound quantitative characterization suitable for PBPK modeling. Thus, we established a coherent framework of VRC’s metabolism assessing relevant aspects of its properties as a substrate and inhibitor to contribute to the elucidation of the complex PK of VRC.

## Figures and Tables

**Figure 1 pharmaceutics-14-00477-f001:**
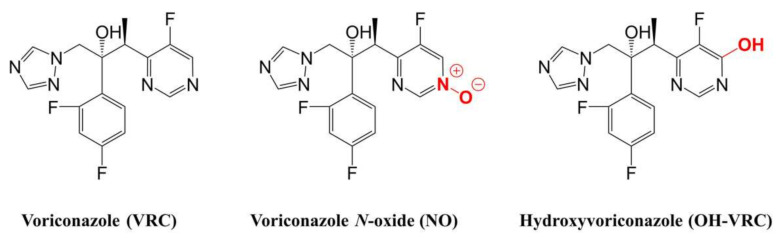
Chemical structures of voriconazole and its two metabolites, voriconazole *N*-oxide and hydroxyvoriconazole.

**Figure 2 pharmaceutics-14-00477-f002:**
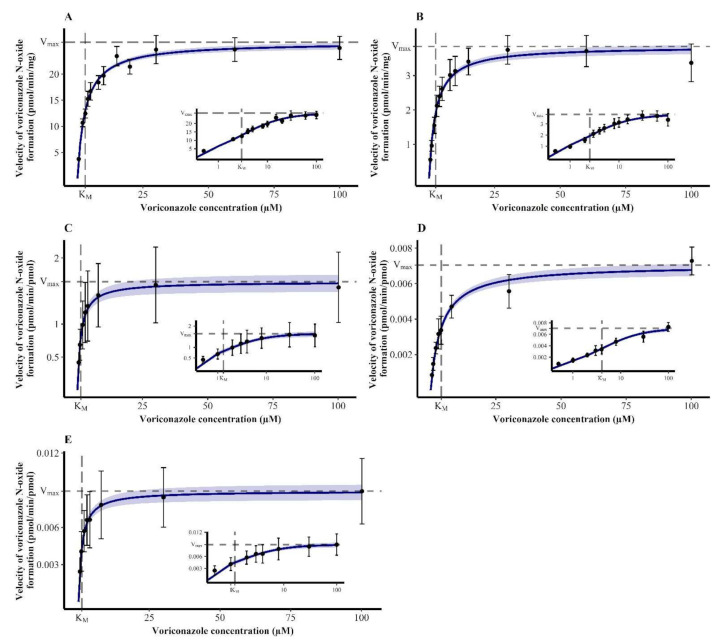
Michaelis-Menten kinetics of the *N*-oxidation of voriconazole in human liver microsomes ((**A**), *n* = 8–9), human intestine microsomes ((**B**), *n* = 9) and recombinant human cytochrome P450 2C19 ((**C**), *n* = 18), 2C9 ((**D**), *n* = 10) and 3A4 ((**E**), *n* = 18). Insets show the respective plot with a log-transformed *x*-axis. Data points—mean reaction velocity; error bars—standard deviation of reaction velocity; solid blue line—estimated enzyme kinetics; shaded area—95% confidence interval of the estimation; dashed lines—estimated Michaelis–Menten constant (K_M_) and maximum reaction velocity (V_max_).

**Figure 3 pharmaceutics-14-00477-f003:**
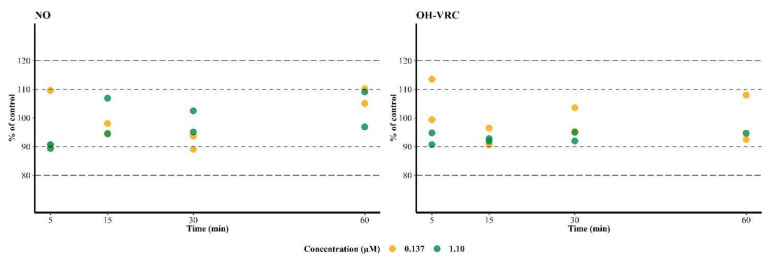
Stability of voriconazole *N*-oxide (NO) and hydroxyvoriconazole (OH–VRC) in human liver microsomes over time at two concentrations (*n* = 1–2 each), presented as the percentage of a control incubation (*n* = 1). Dashed lines—±10% and ±20% deviation from the control.

**Figure 4 pharmaceutics-14-00477-f004:**
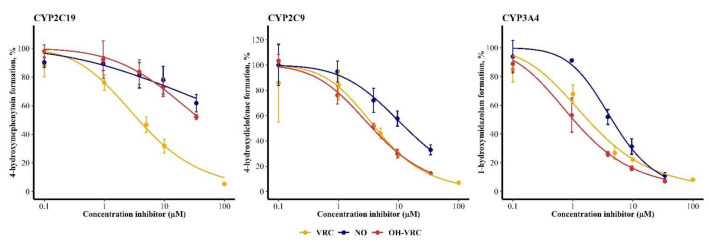
Inhibitory potential of voriconazole (VRC), voriconazole *N*-oxide (NO) and hydroxyvoriconazole (OH–VRC) on CYP2C19, CYP2C9 and CYP3A4 presented as the remaining activity of the respective marker reaction in function of the inhibitor concentration. Data points—mean activity (*n* = 4); error bars—standard deviation of activity; solid lines—estimated relation between activity and inhibitor concentration.

**Figure 5 pharmaceutics-14-00477-f005:**
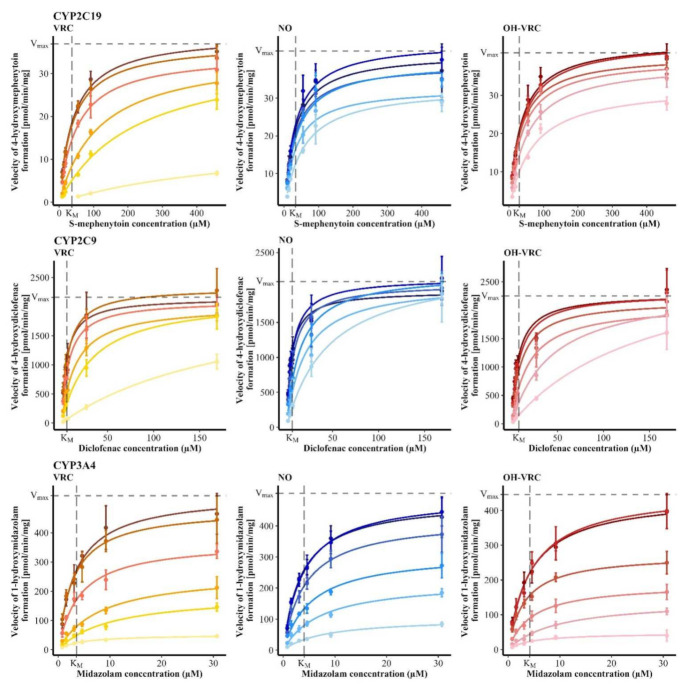
Reaction velocities of 4-hydroxymephenytoin formation by CYP2C19 (**top**), 4-hydroxydiclofenac formation by CYP2C9 (**middle**) and 1-hydroxymidazolam formation by CYP3A4 (**bottom**) in function of the substrate concentration under the influence of increasing concentrations of the inhibitors (increasing inhibitor concentration with decreasing color intensity) voriconazole (VRC, 0, 0.1, 1, 5, 10, 100 µM), voriconazole *N*-oxide (NO, 0, 0.0958, 0.958, 3.83, 9.58 and 34.2 µM) and hydroxyvoriconazole (OH–VRC, 0, 0.0958, 0.958, 3.83, 9.58 and 34.2 µM) in human liver microsomes. Data points—mean reaction velocity (*n* = 2–4); error bars—standard deviation of reaction velocity; solid lines—estimated enzyme kinetics; dashed lines—Michaelis-Menten constant (K_M_) and maximum reaction velocity (V_max_) of the respective uninhibited control.

**Table 1 pharmaceutics-14-00477-t001:** Estimated Michaelis-Menten kinetic parameters, i.e., the Michaelis-Menten constant (K_M_) and maximum reaction velocity (V_max_), and their 95% confidence interval (CI) of the *N*-oxidation of voriconazole determined in human liver microsomes (HLM), human intestine microsomes (HIM) and recombinant human cytochrome P450 enzymes 2C19, 2C9 and 3A4 (rhCYP).

Enzymatic System	K_M_ (95% CI) (µM)	V_max,HLM/HIM_ (95% CI) (pmol/min·mg) or V_max,rhCYP_ (95% CI) (pmol/min·pmol)
HLM	2.98 (2.63–3.33)	26.1 (25.4–26.8)
HIM	2.53 (2.15–2.92)	3.85 (3.70–4.00)
rhCYP2C19	1.31 (0.862–1.75)	1.64 (1.50–1.77)
rhCYP2C9	4.06 (3.32–4.81)	0.00705 (0.00665–0.00744)
rhCYP3A4	1.20 (0.830–1.58)	0.00893 (0.00827–0.00958)

**Table 2 pharmaceutics-14-00477-t002:** Reaction velocities of voriconazole *N*-oxidation in the absence and presence of cytochrome P450 (CYP) enzyme inhibitors, their proportion compared to the uninhibited control incubation and the resulting percentage of inhibition (*n* = 6).

Voriconazole Concentration (µM)	Mean (SD) Reaction Velocity (pmol/min·mg)	% of Uninhibited Control	Mean (SD) % of Uninhibited Control	% Inhibition
Control incubation without inhibitor				
0.5	2.92 (0.288)	100	100	0
1	5.34 (0.574)	100		
2	7.79 (1.05)	100		
3	9.38 (1.11)	100		
Incubation with CYP2C19 inhibitor loratadine				
0.5	1.23 (0.118)	42.0	38.2 (2.64)	61.8
1	2.01 (0.267)	37.7		
2	2.86 (0.351)	36.7		
3	3.40 (0.311)	36.2		
Incubation with CYP2C9 inhibitor sulfaphenazole				
0.5	1.89 (0.124)	64.8	63.8 (2.69)	36.2
1	3.24 (0.431)	60.7		
2	4.88 (0.391)	62.6		
3	6.48 (0.839)	66.9		
Incubation with CYP3A4 inhibitor ketoconazole				
0.5	1.45 (0.140)	49.8	52.4 (4.26)	47.6
1	2.58 (0.248)	48.4		
2	4.17 (0.350)	53.6		
3	5.43 (0.393)	57.9		
Incubation with a mixture of the CYP 2C19, 2C9 and 3A4 inhibitors loratadine, sulfaphenazole and ketoconazole				
0.5	0.249 (0.0151)	8.52	10.8 (2.02)	89.2
1	0.525 (0.0442)	9.83		
2	0.910 (0.111)	11.7		
3	1.23 (0.149)	13.1		

SD—standard deviation.

**Table 3 pharmaceutics-14-00477-t003:** Concentrations of voriconazole (VRC), voriconazole *N*-oxide (NO) and hydroxyvoriconazole (OH–VRC) causing a half maximum inhibitory effect (IC_50_) on the CYP enzymes 2C19, 3A4 and 2C9 without a pre-incubation period (IC_50_) and with a 30 min pre-incubation in the absence (−) or the presence (+) of NADPH.

Enzyme	Inhibitor	IC_50_ (95% Confidence Interval) (µM)	IC_50_ NADPH (−) (95% Confidence Interval) (µM)	IC_50_ Shift NADPH (−)	IC_50_ NADPH (+) (95% Confidence Interval) (µM)	IC_50_ Shift NADPH (+)
CYP2C19	VRC	3.72 (2.85–4.78)	5.59 (4.61–6.74)	0.667	5.02 (4.12–6.08)	0.741
	NO	288 (65.0–31,623)	450 (93.3–339,557)	0.641	320 (52.0–364,870)	0.900
	OH–VRC	41.7 (26.9–89.1)	35.5 (26.8–53.6)	1.17	33.6 (17.0–160)	1.24
CYP2C9	VRC	4.17 (2.54–6.51)	3.31 (2.70–4.01)	1.29	3.16 (2.71–3.67)	1.35
	NO	13.4 (9.90–19.1)	10.1 (8.28–12.4)	1.34	14.9 (11.1–21.3)	0.899
	OH–VRC	3.67 (3.16–4.26)	3.68 (2.91–4.59)	0.997	3.64 (3.01–4.39)	1.01
CYP3A4	VRC	1.76 (1.26–2.36)	2.90 (2.13–3.85)	0.607	2.63 (1.97–3.40)	0.669
	NO	4.48 (3.78–5.29)	6.96 (4.63–10.2)	0.644	2.91 (1.36–6.19)	1.54
	OH–VRC	1.02 (0.796–1.27)	0.579 (0.323–0.966)	1.76	1.57 (1.25–1.95)	0.650

## Data Availability

Data can be received from the authors upon reasonable request.

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
