# Peer review of "Towards the Elucidation of the Pharmacokinetics of Voriconazole: A Quantitative Characterization of Its Metabolism"

_pharmaceutics, 2022, doi:10.3390/pharmaceutics14030477_

Round 1

Reviewer 1 Report

Authors conducted in the present study the characterization of metabolism of voriconazole (VRC), and evaluation of inhibitory effect of VRC and its metabolites on activity of CYP2C19, CYP2C9 and CYP3A4.  Overall, the manuscript is well written, the aim of the study is sound, and the results and conclusions are clear.  The idea of the present study is interesting, but it is necessary to correct several problems before publication in Pharmaceutics.

  1. The authors describe that one of the metabolites of VRC is dihydroxyvoriconazole (line 62).  However, the authors did not investigate the formation of dihydroxyvoriconazole mediated by CYP2C19, CYP2C9 and CYP3A4 in the present study.  The authors should clearly explain why.
  2. In line 523-524, the authors demonstrated that in our investigations OH-VRC was not formed in incubations of HLM, rhCYP or HIM. Despite the fact that OH-VRC is not formed, the authors evaluate inhibitory effect of OH-VRC on activity of each CYP in Figure 5.  Should the results be included in the manuscript?  If so, the authors should clearly explain why.
  3. Based on the results in Figure 5, the authors should show the Km and Vmax values when using the highest concentration inhibitor. The authors demonstrated that inhibition on CYP2C19 by VRC and OH-VRC was determined as a competitive inhibition (line 385-386).  However, when the inhibitory effect of VRC on CYP2C19 activity was confirmed (Fig. 5, CYP2C19, VRC), it seems that the Vmax value also decreased when treated with 100 μM VRC.
  4. In line 438-477, the authors argue that the Km and Vmax values estimated in the present study are valid. The concentrations of methanol and liver microsomes in the present study were 0.5% and 0.2 mg/mL, respectively.  However, these conditions in previous studies are not included in this manuscript.  The authors should include them in the Discussion section.

Author Response

Point-by-point response to the editor’s and reviewers’ comments to pharmaceutics-1494919

“Towards the elucidation of voriconazole pharmacokinetics: a quantitative characterization of its metabolism.” by Schulz et al.

Please note that the editor’s and reviewers’ comments below are given in black, the responses highlighted in blue and the changes to the manuscript (insertions and deletions) are indicated in red. The line numbers given in our “Response” or “Changes to manuscript” sections refer to those in the original manuscript.

Response to editor’s and reviewers’ comments:

Response: Thank you for the overall positive feedback. We highly appreciate the thorough review and the very valuable comments. The constructive feedback stimulated further reflection and we have carefully revised the manuscript according to the reviewers’ suggestions as summarized in the following to further improve the quality of our work.

Reviewer #1:

Authors conducted in the present study the characterization of metabolism of voriconazole (VRC), and evaluation of inhibitory effect of VRC and its metabolites on activity of CYP2C19, CYP2C9 and CYP3A4.  Overall, the manuscript is well written, the aim of the study is sound, and the results and conclusions are clear.  The idea of the present study is interesting, but it is necessary to correct several problems before publication in Pharmaceutics.

Response: We highly appreciate the positive feedback and note the raised points of discussion. The mentioned aspects were carefully discussed and revised as outlined below.

  • The authors describe that one of the metabolites of VRC is dihydroxyvoriconazole (line 62).  However, the authors did not investigate the formation of dihydroxyvoriconazole mediated by CYP2C19, CYP2C9 and CYP3A4 in the present study.  The authors should clearly explain why.

Response: Thank you for making this remark. Based on the knowledge of voriconazole (VRC) metabolism, dihydroxyvoriconazole is a secondary metabolite formed subsequently to the first hydroxylation of VRC at the fluoropyrimidine moiety [1,2]. Consequently, as no formation of hydroxyvoriconazole was observed, the hypothesis emerged that hydroxyvoriconazole is rapidly transformed to dihydroxyvoriconazole and thus not detectable in the respective incubations. To investigate this hypothesis, we incubated hydroxyvoriconazole directly with human liver microsomes with the expectation of depletion over time if indeed dihydroxyvoriconazole is formed. As described in our results (‘3.4 Metabolic stability of voriconazole N-oxide and hydroxyvoriconazole’) this was not the case and hence also the formation of dihydroxyvoriconazole assumed to be based on a different metabolic pathway. We added these findings and the inference to the Discussion.

Changes to manuscript:

Discussion, lines 523-524

However, in our investigations OH-VRC was not formed in incubations of HLM, rhCYP or HIM,. Additionally, the formation of dihydroxyvoriconazole, a secondary metabolite of OH-VRC, was excluded in the experiments indirectly by the observed absence of depletion of OH-VRC in HLM. These findings suggest a different pathway of formation for both metabolites.

  • In line 523-524, the authors demonstrated that in our investigations OH-VRC was not formed in incubations of HLM, rhCYP or HIM. Despite the fact that OH-VRC is not formed, the authors evaluate inhibitory effect of OH-VRC on activity of each CYP in Figure 5. Should the results be included in the manuscript?  If so, the authors should clearly explain why.

Response: The reviewer is correct to point out that a clarification should be made. Although hydroxyvoriconazole (OH-VRC) was found not to be formed by CYP enzymes in the present study, it has been described to occur in humans [2–4]. Hence, the formation might be catalyzed by different (liver) enzymes. We think, that despite its non-identified metabolic pathway in our experiments, the mere presence in vivo is the crucial and valid aspect to also investigate its inhibitory potential on CYP enzymes. Nevertheless, we share the reviewer’s opinion, that the clinical relevance of the observed inhibition must be discussed. Therefore, we state in lines 566-571 of the Discussion, that in particular local liver concentrations of OH-VRC are of interest to evaluate the clinical relevance. As such knowledge is not yet available, we are of the opinion, that it would not be appropriate to withhold the data on OH-VRC inhibitory potential on CYP isoenzymes from the readers.

  • Based on the results in Figure 5, the authors should show the Km and Vmax values when using the highest concentration inhibitor. The authors demonstrated that inhibition on CYP2C19 by VRC and OH-VRC was determined as a competitive inhibition (line 385-386).  However, when the inhibitory effect of VRC on CYP2C19 activity was confirmed (Fig. 5, CYP2C19, VRC), it seems that the Vmax value also decreased when treated with 100 μM VRC.

Response: Thank you for pointing out this detail and we added to the supplementary material a table including all parameter estimates of the respective inhibition model. However, we want to clarify further, that estimation of Ki was based on a modelling approach. Hence, all data points, i.e. reaction velocities in the presence of all respective substrate and inhibitor concentrations, were used simultaneously as input data for the three main models of reversible inhibition (competitive, non-competitive and uncompetitive). All models were fit to all data points, as described by the equations 10 to 12. For the determination of the best model fit, and thus the most appropriate model with the inference of the presence of competitive, non-competitive or uncompetitive inhibition, Akaike’s information criteria [5] was applied. As a result, the kinetic parameters, i.e. Ki, KM and Vmax, of the identified model were reported. Overall, the graphical depiction might be misleading as the reaction velocity, at a VRC concentration of 100 µM, is only investigated up to a S-mephenytoin concentration of 458 µM and is likely to increase further at higher substrate concentrations.

Changes to manuscript:

Results, lines 390-395

All three substances inhibited CYP2C9 in a competitive manner and yielded Ki of 2.57 µM (2.16 – 3.14 µM) for VRC, 5.47 µM (4.32 – 7.00 µM) for NO and 2.80 µM (2.20 – 3.61 µM) for OH-VRC. The inhibition of VRC, NO and OH-VRC on CYP3A4 was noncompetitive and resulted in Ki of 2.75 µM (2.35 – 3.22 µM), 5.24 µM (4.68 – 5.86 µM) and 2.53 µM (2.24 – 2.87 µM), respectively (Figure 5). The simultaneously estimated KM and Vmax values are presented in Table S4.

Back matter, lines 608- 619

Supplementary Materials: The following are available online at www.mdpi.com/xxx/s1, Supplementary Section S1. Marker reactions for CYP2C19, CYP2C9 and CYP3A4; Table S1. Physiological parameters taken from literature to perform in vitro in vivo extrapolation; Table S2. Specificity of the CYP2C19, CYP2C9 and CYP3A4 inhibitors loratadine, sulfaphenazole and ketoconazole on the marker reactions of the respective enzymes determined as remaining reaction velocity compared to a control incubation without inhibitor in human liver microsomes; Table S3. Michaelis-Menten kinetic parameters for the marker reactions of CYP2C19, CYP2C9 and CYP3A4; Table S4. Type of inhibition caused by voriconazole (VRC), voriconazole N-oxide (NO) and hydroxyvoriconazole (OH-VRC) on the CYP isoenzymes specific reactions of S-Mephenytoin 4-hydroxylation (2C19), diclofenac 4‑hdyroxylation (2C9) and midazolam 1‑hydroxylation (3A4) and the associated inhibitory constants (Ki) as well as the Michaelis-Menten constants (KM) and maximum reaction velocities (Vmax); Figure S1. Inhibitory potential of voriconazole, voriconazole N-oxide and hydroxyvoriconazole on CYP2C19, CYP2C9 and CYP3A4 without a pre-incubation period of human liver microsomes and inhibitor and with a pre-incubation period of 30 min in the absence and presence of NADPH re-generating system. Presented is the remaining activity of the respective marker reaction in dependence of the inhibitor concentration.

Supplementary material, line 66

Table S4: Type of inhibition caused by voriconazole (VRC), voriconazole N-oxide (NO) and hydroxyvoriconazole (OH-VRC) on the CYP isoenzyme specific reactions of S-Mephenytoin 4-hydroxylation (2C19), diclofenac 4‑hdyroxylation (2C9) and midazolam 1-hydroxylation (3A4) and the associated inhibitory constants (Ki) as well as the Michaelis-Menten constants (KM) and maximum reaction velocities (Vmax).

Enzyme

Inhibitor

Type of reversible inhibition

Ki (95% CI)

[µM]

KM (95% CI)

[µM]

Vmax (95% CI)

[pmol/min∙mg]

CYP2C19

VRC

Competitive

1.90

(1.70–2.12)

37.0

(34.3–40.0)

36.9

(36.1–37.8)

NO

Non-competitive

58.6

(46.8–75.2)

33.1

(29.8–36.7)

42.4

(40.8–44.0)

OH-VRC

Competitive

11.6

(9.65–14.0)

29.7

(27.3–32.2)

41.1

(40.1–42.1)

CYP2C9

VRC

Competitive

2.57

(2.16–3.14)

5.80

(5.14–6.52)

2159

(2089–2230)

NO

Competitive

5.47

(4.32–7.00)

6.32

(5.60–7.13)

2084

(2016–2154)

OH-VRC

Competitive

2.80

(2.20–3.61)

7.93

(6.81–9.23)

2251

(2158–2346)

CYP3A4

VRC

Non-competitive

2.75

(2.35–3.22)

3.52

(3.05–4.05)

527

(500–554)

NO

Non-competitive

5.24

(4.68–5.86)

4.05

(3.65–4.48)

505

(486–525)

OH-VRC

Non-competitive

2.53

(2.24–2.87)

4.16

(3.71–4.67)

446

(427–466)

CI: confidence interval

  • In line 438-477, the authors argue that the Km and Vmax values estimated in the present study are valid. The concentrations of methanol and liver microsomes in the present study were 0.5% and 0.2 mg/mL, respectively.  However, these conditions in previous studies are not included in this manuscript.  The authors should include them in the Discussion section.

Response: Thank you for pointing this out. Indeed, we wanted to include these details in the discussion, however the publications by Hyland et al. and Murayama et al. [6,7] are ambiguous for the content of organic solvent in their incubations. Solubility of VRC in water is reported as 0.0978 mg/mL (https://pubchem.ncbi.nlm.nih.gov/compound/Voriconazole#section=Solubility). However, as for kinetic investigations VRC concentrations of up to 2500 and 5000 µM were reported by the authors, organic solvent concentrations must have been either very high or VRC precipitated in the final incubation. In both cases, kinetic parameters are likely to be distorted. Regarding the microsomal protein content, Hyland et al. did not report a concentration for kinetic investigations. However, when the authors investigated VRC metabolism in CYP2C19 poor metabolizer liver microsomes, they used 1 mg/mL. Murayama et al. reported a microsome concentration of 0.5 mg/mL. Thus, we included this information in our manuscript.

Changes to manuscript:

Discussion, lines 466-470

Fourth, the usage of high HLM (>0.5 mg/mL) or rhCYP concentrations, e.g. 0.5 or 1 mg/mL [31,33], potentially increases nonspecific protein binding of the substrate to the protein and thus reduces the unbound fraction of substrate [71]. Therefore, for unbiased kinetic determinations the unbound substrate concentration should be used for determinations of Michaelis-Menten kinetics [69].

Reviewer #2:

In the present manuscript, authors have carried out the in vitro experiments to understand the Phase I CYP mediated metabolism of Voriconazole and the CYP inhibition potential of VRC and its metabolites. The manuscript has been well written in general, specific comments are below:

Response: Thank you for the overall positive response. We highly appreciate the thorough review and the very valuable comments from the reviewer.

  • Section 2.4 - Authors claim hydroxy-VRC formation was CYP and FMO independent. What experiments were done to understand FMO mdeiated metabolism?

Response: Human liver microsomes (HLM) contain, besides CYP isoenzymes, also flavin containing monooxygenases (FMO). FMO have a comparable mode of action as CYP enzymes and were assumed to be active in the current experimental set-up [8]. Hence, we concluded, that the absence of OH-VRC in HLM incubations suggested a CYP and FMO independent metabolic pathway.

  • Line 173 - 'analyzed in dependence of the added substrate concentration' - what does in dependence imply here? Is this a typo?

Response: Thank you for pointing out this ambiguous phrasing. We adapted this part in the manuscript.

Changes to manuscript:

Materials and Methods, lines 173

The reaction velocity of metabolite formation (V) was analyzed in dependence function of the added substrate concentration ([S] in [µM]) and fitted to using the Michaelis-Menten equation (Eq. 2).

  • Why was the HIM clearance not scaled? Are the ISEF expts done with markers relevant for HIM as well?

Response: Thank you for indicating this point of discussion. Although a “well-stirred gut model” exists, it is not as well established as the “well-stirred liver model” (equation 8). In particular the term of blood flow (Q) has been controversially discussed, as the drug does not reach the metabolic enzymes via the blood flow, but from the lumen. Consequently, more complex models have been proposed. However, those require assumptions on the permeability through the enterocyte and villous blood flow [9]. Nevertheless, we agree with the reviewer, that the fraction of (intrinsic) clearance of the intestine of total VRC metabolic clearance is of high interest. However, to cope with the complexity of the described processes the application of a physiologically-based pharmacokinetic (PBPK) model presents a promising next step of the current work. Based on the in silico gained knowledge, further experiments such as the determination of CYP isoenzyme contributions in the intestines should be considered.

Discussion, lines 516-518

Thus, the estimated hepatic clearances of 127 - 266 mL/min as derived from the here presented IVIVE represents a good approximation, if considering that only one metabolic pathway, VRC N-oxidation, was taken into account. The extrapolation from HIM and the determination of the fraction of intestinal (intrinsic) clearance of the total VRC metabolic clearance is a promising next step of the current work by the application of PBPK modeling.

  • How was Ki determined?

Response: In our work, we determined Ki by investigations of the reaction velocity of CYP isoenzyme marker reactions, i.e. S-mephenytoin 4-hydroxylation (CYP2C19), diclofenac 4-hydroxylation (CYP2C9) and midazolam 1-hydroxylation (CYP3A4) in the presence of VRC, NO and OH-VRC. Overall, reaction velocities were determined at six substrate and six inhibitor concentrations (incl. 0 µM, =36 combinations), all combinations were performed in duplicates. Afterwards, the three main models of reversible inhibition, i.e. competitive, non-competitive and uncompetitive (equations 10 to 12), were fitted to all data using a nonlinear regression analyzes. Subsequently, the determination of the best model fit was assessed by Akaike’s information criteria [5] and the respective parameter estimates, i.e. Ki, KM and Vmax reported (lines 239-250 in the manuscript). Please see also the response to the third remark of Reviewer #1 and the respective changes to the manuscript.

Reviewer #3:

Excellent written article. In my opinion it can be published as it is. 

Response: Thank you for the very positive response, we appreciate the feedback.

References

  1. Schulz, J.; Kluwe, F.; Mikus, G.; Michelet, R.; Kloft, C. Novel Insights into the Complex Pharmacokinetics of Voriconazole: A Review of Its Metabolism. Drug Metabolism Reviews 2019, 51, 247–265, doi:10.1080/03602532.2019.1632888.
  2. Roffey, S.J.; Cole, S.; Comby, P.; Gibson, D.; Jezequel, S.G.; Nedderman, A.N.R.; Smith, D.A.; Walker, D.K.; Wood, N. THE DISPOSITION OF VORICONAZOLE IN MOUSE, RAT, RABBIT, GUINEA PIG, DOG, AND HUMAN. Drug Metabolism and Disposition 2003, 31, 731–741, doi:10.1124/dmd.31.6.731.
  3. Geist, M.J.P.; Egerer, G.; Burhenne, J.; Riedel, K.-D.; Weiss, J.; Mikus, G. Steady-State Pharmacokinetics and Metabolism of Voriconazole in Patients. Journal of Antimicrobial Chemotherapy 2013, 68, 2592–2599, doi:10.1093/jac/dkt229.
  4. Scholz, I.; Oberwittler, H.; Riedel, K.-D.; Burhenne, J.; Weiss, J.; Haefeli, W.E.; Mikus, G. Pharmacokinetics, Metabolism and Bioavailability of the Triazole Antifungal Agent Voriconazole in Relation to CYP2C19 Genotype. British Journal of Clinical Pharmacology 2009, 68, 906–915, doi:10.1111/j.1365-2125.2009.03534.x.
  5. Akaike, H. A New Look at the Statistical Model Identification. IEEE Transactions on Automatic Control 1974, 19, doi:10.1109/TAC.1974.1100705.
  6. Murayama, N.; Imai, N.; Nakane, T.; Shimizu, M.; Yamazaki, H. Roles of CYP3A4 and CYP2C19 in Methyl Hydroxylated and N-Oxidized Metabolite Formation from Voriconazole, a New Anti-Fungal Agent, in Human Liver Microsomes. Biochemical Pharmacology 2007, 73, 2020–2026, doi:10.1016/j.bcp.2007.03.012.
  7. Hyland, R.; Jones, B.C.; Smith, D.A. Identification of the Cytochrome P450 Enzymes Involved in the N -Oxidation of Voriconazole. Drug Metabolism and Disposition 2003, 31, 540–547, doi:10.1124/dmd.31.5.540.
  8. Yanni, S.B.; Annaert, P.P.; Augustijns, P.; Bridges, A.; Gao, Y.; Benjamin, D.K.; Thakker, D.R. Role of Flavin-Containing Monooxygenase in Oxidative Metabolism of Voriconazole by Human Liver Microsomes. Drug Metabolism and Disposition 2008, 36, 1119–1125, doi:10.1124/dmd.107.019646.
  9. Yang, J.; Jamei, M.; Yeo, K.; Tucker, G.; Rostami-Hodjegan, A. Prediction of Intestinal First-Pass Drug Metabolism. Current Drug Metabolism 2007, 8, 676–684, doi:10.2174/138920007782109733.

Reviewer 2 Report

In the present manuscript, authors have carried out the in vitro experiments to understand the Phase I CYP mediated metabolism of Voriconazole and the CYP inhibition potential of VRC and its metabolites. The manuscript has been well written in general, specific comments are below: 

-Section 2.4 - Authors claim hydroxy-VRC formation was CYP and FMO independent. What experiments were done to understand FMO mdeiated metabolism ? 

-Line 173 - 'analyzed in dependence of the added substrate concentration' - what does in dependence imply here? Is this a typo ?

-Why was the HIM clearance not scaled ? Are the ISEF expts done with markers relevant for HIM as well ?

-How was Ki determined ?

Author Response

(The authors gave the same response as above.)

Reviewer 3 Report

Excellent written article. In my opinion it can be published as it is. 

Author Response

(The authors gave the same response as above.)

Round 2

Reviewer 1 Report

I thought it was appropriate for the revised manuscript to be published in the Journal "Pharmaceutics".